# Persistent Planar Tetracoordinate Carbon in Global Minima Structures of Silicon-Carbon Clusters

**Luis Leyva-Parra** [1,2], **Diego Inostroza** [1,2], **Osvaldo Yañez** [3,4], **Julio César Cruz** [5], **Jorge Garza** [5], **Víctor García** [6,*] **and William Tiznado** [1,*]

[1] Computational and Theoretical Chemistry Group, Departamento de Ciencias Química, Facultad de Ciencias Exactas, Universidad Andres Bello, República 498, Santiago 8370251, Chile; l.leyvaparra@uandresbello.edu (L.L.-P.); dinostro92@gmail.com (D.I.)

[2] Facultad de Ciencias Exactas, Programa de Doctorado en Fisicoquímica Molecular, Universidad Andres Bello, Santiago 8370251, Chile

[3] Center of New Drugs for Hypertension (CENDHY), Santiago 8380494, Chile; osvyanezosses@gmail.com

[4] Department of Pharmaceutical Science and Technology, School of Chemical and Pharmaceutical Sciences, Universidad de Chile, Santiago 8380494, Chile

[5] Departamento de Química, División de Ciencias Básicas e Ingeniería, Universidad Autónoma Metropolitana-Iztapalapa, San Rafael Atlixco 186, Col. Vicentina, Iztapalapa, Mexico City 09340, Mexico; julio.cruz088@gmail.com (J.C.C.); jgo@xanum.uam.mx (J.G.)

[6] Facultad de Química e Ingeniería Química, Universidad Nacional Mayor de San Marcos, Lima 15004, Peru

[*] Correspondence: victor.garcia@unmsm.edu.pe (V.G.); wtiznado@unab.cl (W.T.)

**Abstract:** Recently, we reported a series of global minima whose structures consist of carbon rings decorated with heavier group 14 elements. Interestingly, these structures feature planar tetracoordinate carbons (ptCs) and result from the replacement of five or six protons ($H^+$) from the cyclopentadienyl anion ($C_5H_5^-$) or the pentalene dianion ($C_8H_6^{2-}$) by three or four $E^{2+}$ dications (E = Si–Pb), respectively. The silicon derivatives of these series are the $Si_3C_5$ and $Si_4C_8$ clusters. Here we show that ptC persists in some clusters with an equivalent number of C and Si atoms, i.e., $Si_5C_5$, $Si_8C_8$, and $Si_9C_9$. In all these species, the ptC is embedded in a pentagonal $C_5$ ring and participates in a three-center, two-electron (3c-2e) Si-ptC-Si σ-bond. Furthermore, these clusters are π-aromatic species according to chemical bonding analysis and magnetic criteria.

**Keywords:** planar tetracoordinate carbon; clusters; global minima; DFT computations; chemical bonding analysis; aromaticity

## 1. Introduction

Chemists have a great interest in predicting new chemical entities with exotic non-classical structures. Planar hypercoordinate carbon (phC) atoms, i.e., molecules containing carbon atoms linked to four or more ligands in-plane, are particularly puzzling. The phCs violate the well-established rule of van't Hoff and Le Bel (regarding the concept of tetrahedral four-coordinate carbon); thus, at the beginning, they were considered experimentally inaccessible. However, in 1968, Monkhorst evaluated, theoretically, methane stereomutation through a planar tetracoordinate carbon (ptC) transition state [1]. Two years later, Hoffmann and co-workers proposed different approaches to stabilize a ptC with the modest purpose of achieving a thermally accessible transition state for a racemization process [2]. These pioneering works inspired different studies that finally allowed the identification of viable ptC compounds [3–7].

In the last 50 years, significant progress has been made in synthesizing and "in silico" proposals of numerous ptC compounds [3–5,8–11]. Even more recently, the chemistry of the family was extended to species in which the carbon coordination number is higher than four (penta [12–22] and hexacoordinate [23–26]). In many of these systems, the formation

of delocalized bonds plays a decisive role in their stability. For instance, the experimentally detected ptC $CAl_4^{2-}$ cluster exhibits doubly σ- and π-aromatic character [10,11].

In 2019, some of the current authors designed a series of ptC global minima composed of carbon and heavier group 14 elements (Si–Pb) [7]. These clusters were obtained by replacing five or six protons ($H^+$) from the cyclopentadienyl anion ($C_5H_5^-$) or pentalene dianion ($C_8H_6^{2-}$) by three or six $E^{2+}$ dications (E = Si–Pb), respectively. In these clusters, the π-aromatic circuits of the parent aromatic hydrocarbons are preserved. The global minima structures of the clusters $Si_3C_5$, $Ge_3C_5$, $Si_4C_8$, and $Ge_4C_8$ contain one or two ptCs (see Figure 1). Chemical bonding analysis suggests that these clusters are globally π-aromatic and locally σ-aromatic, where the local aromaticity is due to the E-ptC-E 3c-2e σ-bond. It is important to note that a similar strategy has successfully allowed other aromatic hydrocarbon derivatives with ptCs [6,22,27,28].

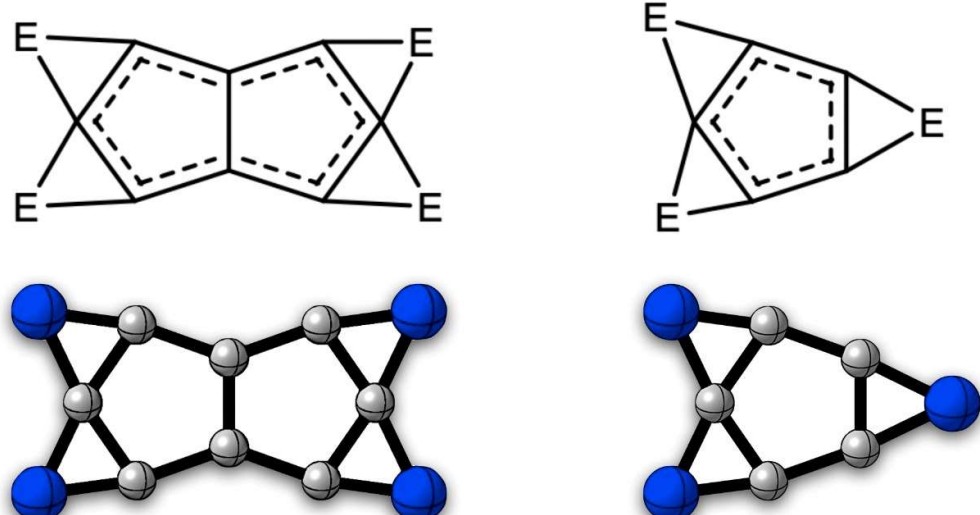

**Figure 1.** $E_3C_5$ and $E_4C_8$ (E = Si–Pb) systems proposed in reference [7], when E = Si and Ge structures correspond to the global minima (see bottom of the figure).

The current work highlights that ptCs are more common than one would think in Si-C clusters. For example, in the $Si_nC_n$ combinations, when *n* = 5, 8, and 9, the global minima contain ptCs. This feature was not perceived in the works that reported these structures [29–31], which has motivated us to highlight it here, and analyze these species' chemical bonding patterns. Remarkably, silicon carbide (SiC) grains are major dust components in carbon-rich stars of the asymptotic giant branch [29]; and there is evidence that combinations of carbon and silicon with suitable stoichiometries favor the formation of two-dimensional materials with the consequent hypercoordination of C and Si [32,33]. Thus, some of these clusters could be involved in these materials' formation or fragmentation processes. On the other hand, the ptC in the studied clusters is embedded in a pentagonal $C_5$ ring. This evidence leads us to propose that any polycyclic hydrocarbon, with pentagonal aromatic rings, can be transformed into Si-C clusters favoring the formation of ptCs.

## 2. Computational Details

The potential energy surface of the species $Si_5C_5$, $Si_8C_8$, and $Si_9C_9$ was explored using the AUTOMATON program [34,35]. Geometry optimizations were performed at the PBE0 [36]/def2–TZVP [37] level. Low-lying isomers were reoptimized at the PBE0–D3 [38]/def2–TZVP level, where dispersion is included in the functional, to identify possible effects on the relative energies. Vibrational frequencies were evaluated at the same level to confirm the optimized structures as true minima on their potential energy surface. These computations were performed using the Gaussian16 program [39].

We computed current densities at PBE0/def2-TZVP level using the GIMIC program [40,41], which employs the gauge-including atomic orbital (GIAO) [42] method. The calculations consider a magnetic field directed along the *z*-axis, i.e., perpendicular to the molecular plane. Note that the unit for current susceptibility is $nA \cdot T^{-1}$, and the results are, therefore, independent of the magnitude of the magnetic field. We prepared vector plots of the current density in a plane placed 0.5 Å above the molecular plane. In our analysis, diatropic (aromatic) and paratropic (antiaromatic) currents circle clockwise and counterclockwise, respectively. To visualize current pathways, we used Paraview 5.10.0 software [43,44]. The ring current strengths (RCS), a measure of the net current intensity around a molecular ring of interest, were obtained after considering different integration planes. The integration planes correspond to cut-off planes perpendicular to the chosen bonds of the annular moiety and extend horizontally for 3.6 Å along the ring's plane, with 2.6 Å above and below the ring. For the integration of the current density passing through an integration plane, GIMIC uses the two-dimensional Gauss–Lobatto algorithm [41,45]. For the RCS, positive (diatropic) and negative (paratropic) signs correspond to the aromatic and anti-aromatic molecules, respectively. RCS values close to zero suggest non-aromatic character [46].

To gain insights about chemical bonding, we used different methods: Wiberg bond indices (WBI) [47], natural population analysis (NPA) [48], and the adaptive natural density partitioning (AdNDP) method [49,50]. These approaches are based on the natural bonding orbital (NBO) method and were performed at the PBE0/def2–TZVP level. The WBI and NPA were computed with the NBO 6.0 code [51], and [41] the AdNDP analysis was performed using Multiwfn 3.8 [52]. The molecular structure and AdNDP results were visualized using CYLview 2.0 [53] and VMD 1.9.3 [54], respectively.

## 3. Results and Discussion

Figure 2 reports the global minima structures of the clusters analyzed in this study: $Si_5C_5$, $Si_8C_8$, and $Si_9C_9$. We can confirm from our searches using the AUTOMATON program that the identified lowest energy structures are the same as those reported in references [29–31]. Note that the dispersion inclusion in the calculations, obtained by Grimme's method (D3), has no significant influence on the relative energy of the lower energy isomers (see Figures S1–S3, and the Cartesian coordinates reported in Table S1). For $Si_5C_5$, the C atoms form a pentagonal ring resembling the cyclopentadienyl anion. On the other hand, for $Si_8C_8$ and $Si_9C_9$, more indene-like carbon structures are formed. Nonetheless, one Si replaces a C for the former to close the six-membered ring.

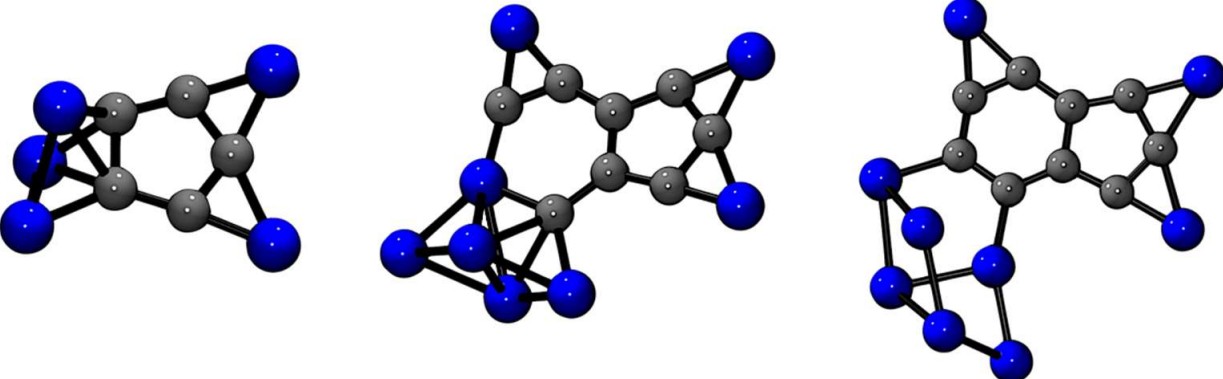

**Figure 2.** Structures of the $Si_nC_n$ global minimum (n = 5, 8, 9) as reported in references [29–31] and confirmed in this work.

What about chemical bonding in these species? In general, we are interested in looking at the presence of both the 3c-2e bond of Si-ptC-Si and the $\pi$-aromatic circuit around the planar carbonaceous moiety. The NPA analysis predicts a negative partial charge distribution on the C and positive on the Si, agreeing with the differences in C/Si electronegativities. However, the most negative charges (about $-0.9\ |e|$) are on the Si's surrounding the ptC (see Figure S4). These results agree with the design model for these species, where protons are replaced by $Si^{2+}$ dication.

Figure 3 shows the chemical bonding interpretation for the $Si_5C_5$ cluster according to AdNDP analysis. There are five lone pairs (one on each Si), five 2c-2e C-C $\sigma$-bonds connecting the $C_5$ ring (similar to the cyclopentadienyl anion $C_5H_5^-$). The Si's on the plane are connected to two C's (ptC neighbors) of the $C_5$ ring by 2c-2e C-Si $\sigma$-bonds and participate in a 3c-2e Si-ptC-Si $\sigma$-bond. It also recovered three $\pi$-bonds distributed on the planar $C_5Si_2$ fragment, suggesting the possibility of aromaticity according to Hückel's rule. The out-of-plane $Si_3$ fragment shares one edge of the $C_5$ pentagon; it exhibits three 3c-2e $\sigma$-bonds and one 2c-2e $\pi$-bond. These results show that $Si_5C_5$ retains much of the $C_5H_5^-$ $\pi$-aromatic features. It also exhibits the multicentric Si-ptC-Si 3c-2e $\sigma$-bond, a feature highlighted as a local aromaticity and stabilizing factor in these species [2,6,7,22]. This chemical bonding picture entirely agrees with the Wiberg bond indexes (WBI) analysis. Figure S4 shows that five C-C single bonds (with some contribution of double bonds) join the $C_5$ ring with WBI values between 1.02 and 1.33 (higher than a single bond), the lowest being for the C-C bonded to the out-of-plane $Si_3$ fragment. The in-plane Si-C and Si-ptC bonds have a WBI value of 1.03 and 0.51, in complete agreement with the single and multicentric bonds recovered by AdNDP analysis.

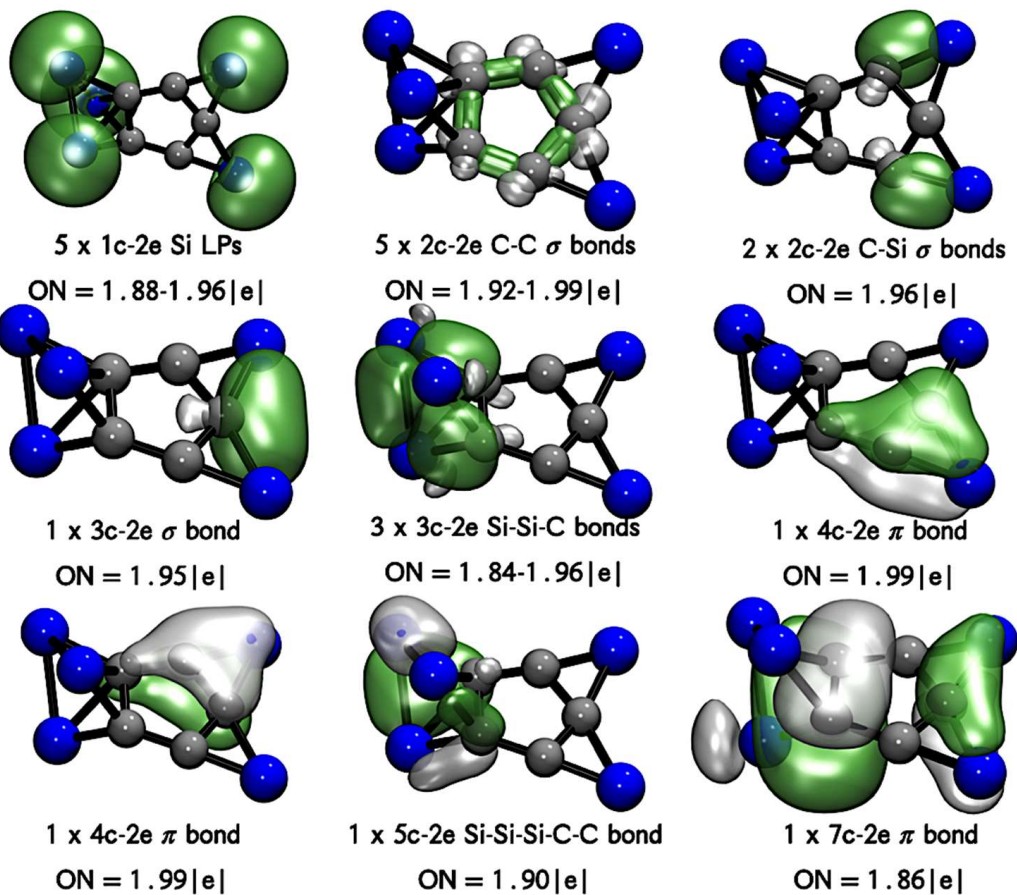

**Figure 3.** AdNDP bonding pattern of cluster $Si_5C_5$. Occupation numbers (ONs) are shown.

In the case of $Si_8C_8$ and $Si_9C_9$ (Figures 4 and 5), AdNDP predicts that both have a pentagonal $C_5$ ring connected by 2c-2e C-C σ-bonds. The $C_5$ pentagon of $Si_8C_8$ is connected to the chain -C-C-Si-C- by two neighboring vertices. Where 2c-2e σ-bonds link carbons, the Si of this chain forms multicentric bonds with the out-of-plane $Si_4$ fragment and the neighboring C's. In the case of $Si_9C_9$, the pentagonal ring is connected to a $C_4$ chain, forming a bicycle analogous to the indene, where 2c-2e σ-bonds connect this bicycle. In both systems, two in-plane Si's are coordinating the ptC (embedded in the $C_5$ pentagon), forming part of a 3c-2e Si-ptC-Si σ-bond, in addition to 2c-2e σ-bonds with the C's neighboring the ptC. In addition, $Si_9C_9$ has one of its Si as a bridge (single) on one side of the $C_6$ hexagon, linked to the C's by Si-C 2c-2e σ-bonds. All in-plane bridged silicon have a lone pair of electrons, in agreement with the design criteria for these species [2,6,7,22]. Both clusters also exhibit π-bonds, six in both cases, which could be associated with an aromatic character. Interestingly, as in $Si_5C_5$, the π orbitals are located around the $SiC_2$ fragments (bridged silicon). This bonding feature is also in complete agreement with the initial design strategy of these systems, where it is advisable to use ligands that can receive electrons in their $p_z$ orbitals, and thus participate in the π-circuit and aromaticity when it is favored [2,6,7,22]. As with the $Si_5C_5$ cluster, the chemical bonding description provided by AdNDP is in complete agreement with the WBI analysis (see Figure S4). WBI predicts bond orders for the C-C bonds slightly higher than the single bonds (higher than 1.15). For $Si_8C_8$, the Si that closes the hexagonal ring and which is annealed to the pentagonal ring, has a bond order of 0.82 and 0.84, agreeing with the detection of multicentric bonds according to AdNDP. Besides, Si's bridged on the hexagonal $C_6$ ring of $Si_9C_9$ have Si-C bond orders of 0.75 and 0.81 in agreement with the detection of two S-C single bonds by AdNDP.

We will now analyze the magnetic response to an external magnetic field applied perpendicular to the Cn plane. This analysis aims to identify patterns related to the (anti) aromaticity phenomenon, i.e., the presence of (diatropic) paratropic ring currents, and to evaluate their intensity (ring current strength, RCS). For reference, the ring current circuits of benzene have been analyzed at the level used in this work. As can be seen from the results (Figure S5), benzene has a paratropic ring current of medium intensity ($-3.5$ nA.T$^{-1}$) inside the $C_6$ ring (from σ-electrons). In addition, there is an intense diatropic ring current (15.3 nA.T$^{-1}$) around the $C_6$ ring, giving an appreciable diatropic resultant of 11.8 nA.T$^{-1}$, characterizing this aromatic molecule par excellence [40,55–58].

In this study, the RCS has been evaluated using integration planes for this purpose, placed bisecting different bonds. In addition, profiles of these integration planes have been analyzed to facilitate the dissection of the different contributions: local, semi local, and global. The analysis used to identify the different ring current circuits has been performed according to the strategy initially proposed in references [46,59], which consists of establishing equations that account for the couplings of the different current flows. Note that this strategy has been used satisfactorily in our group in different recent works [60,61]. The integration planes used and the RCS values (total and partial: profiles) are reported in Figures S3–S5 in the Supplementary Material. From the results (summarized in Figure 6), we wish to highlight first the presence of a moderately intense local diatropic current around the ptC, of 4.2, 4.8, and 5.6 nAT$^{-1}$, for $Si_5C_5$, $Si_8C_8$, and $Si_9C_9$, respectively. The presence of this ring current agrees with the AdNDP analysis, which detects a multicentric Si–ptC–Si σ-bond, thus providing support for the presence of a stabilizing local aromatic contribution. Next, all local $C_5$ and $C_6$ rings have a low-intensity inner paratropic current (between $-5.4$ and $-1.3$ nA.T$^{-1}$), which would be σ in nature, analogous to that exhibited by aromatic hydrocarbons (see the case of benzene mentioned above). $Si_5C_5$ exhibits a diatropic current involving the $C_2$-ptC-$Si_2$ fragment, but not the entire $C_5$ pentagon. This arises from the π-electron cloud involving the Si bridged to the $C_5$ ring.

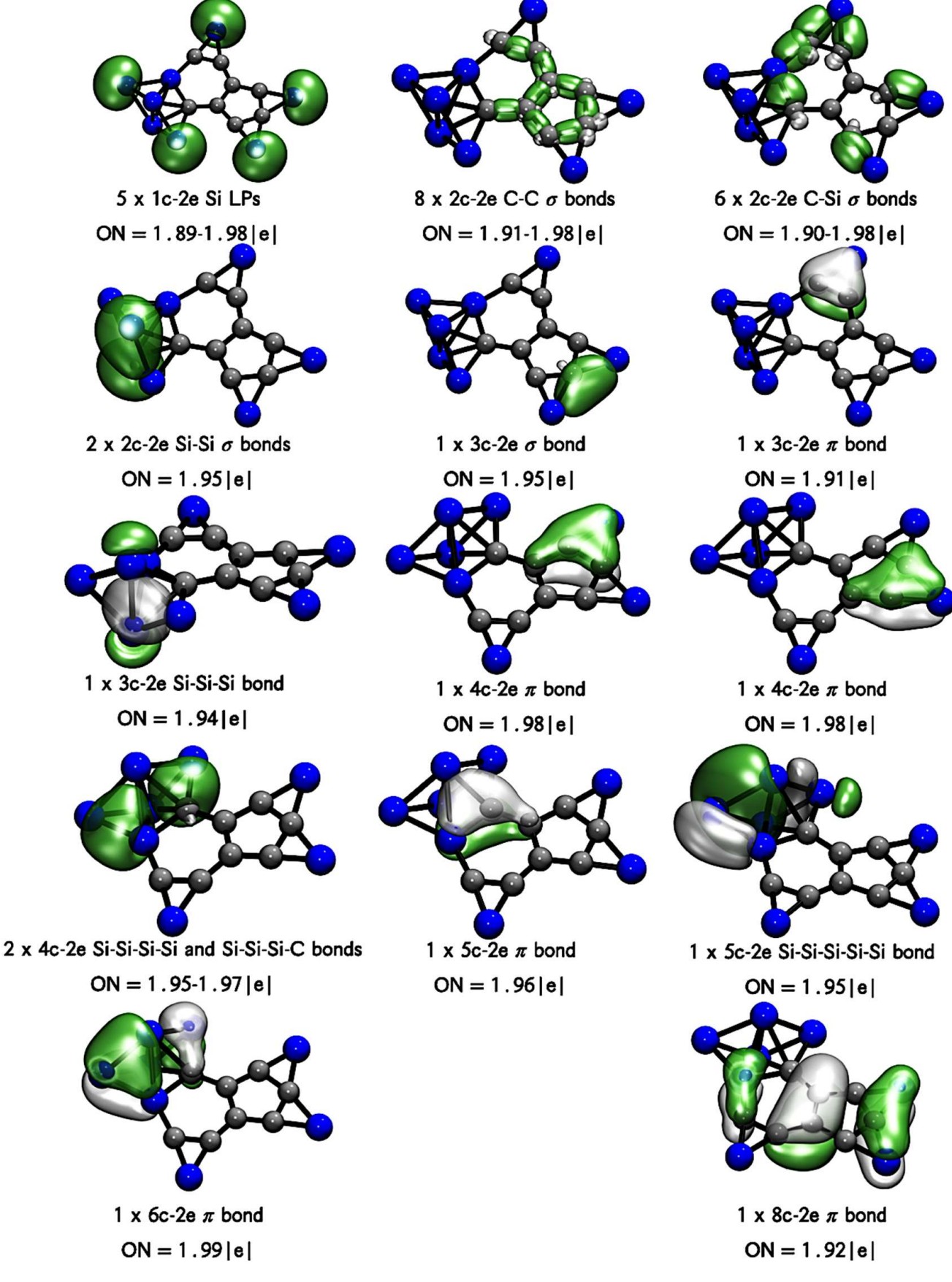

**Figure 4.** AdNDP bonding pattern of cluster $Si_8C_8$. Occupation numbers (ONs) are shown.

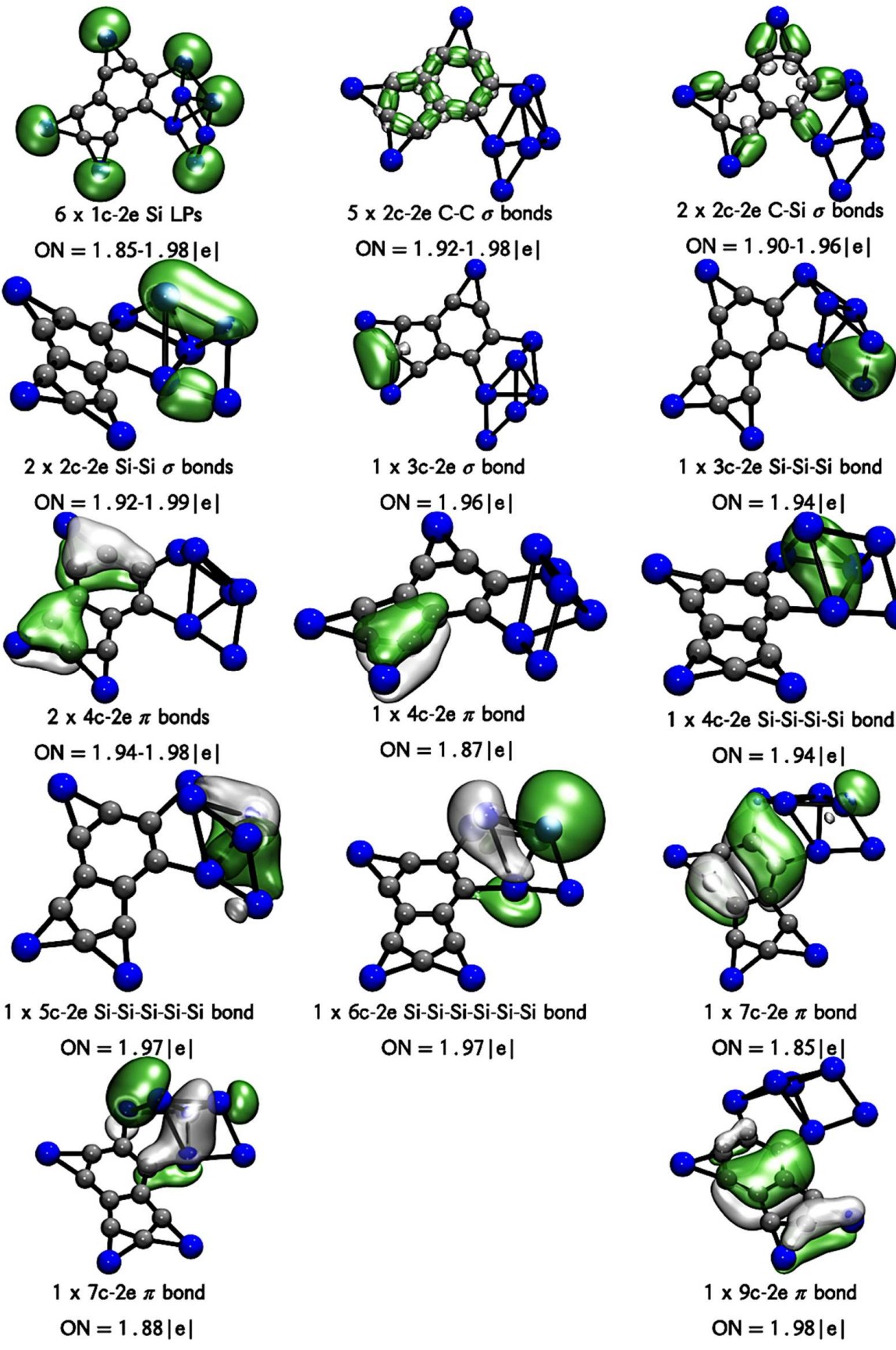

**Figure 5.** AdNDP bonding pattern of cluster $Si_9C_9$. Occupation numbers (ONs) are shown.

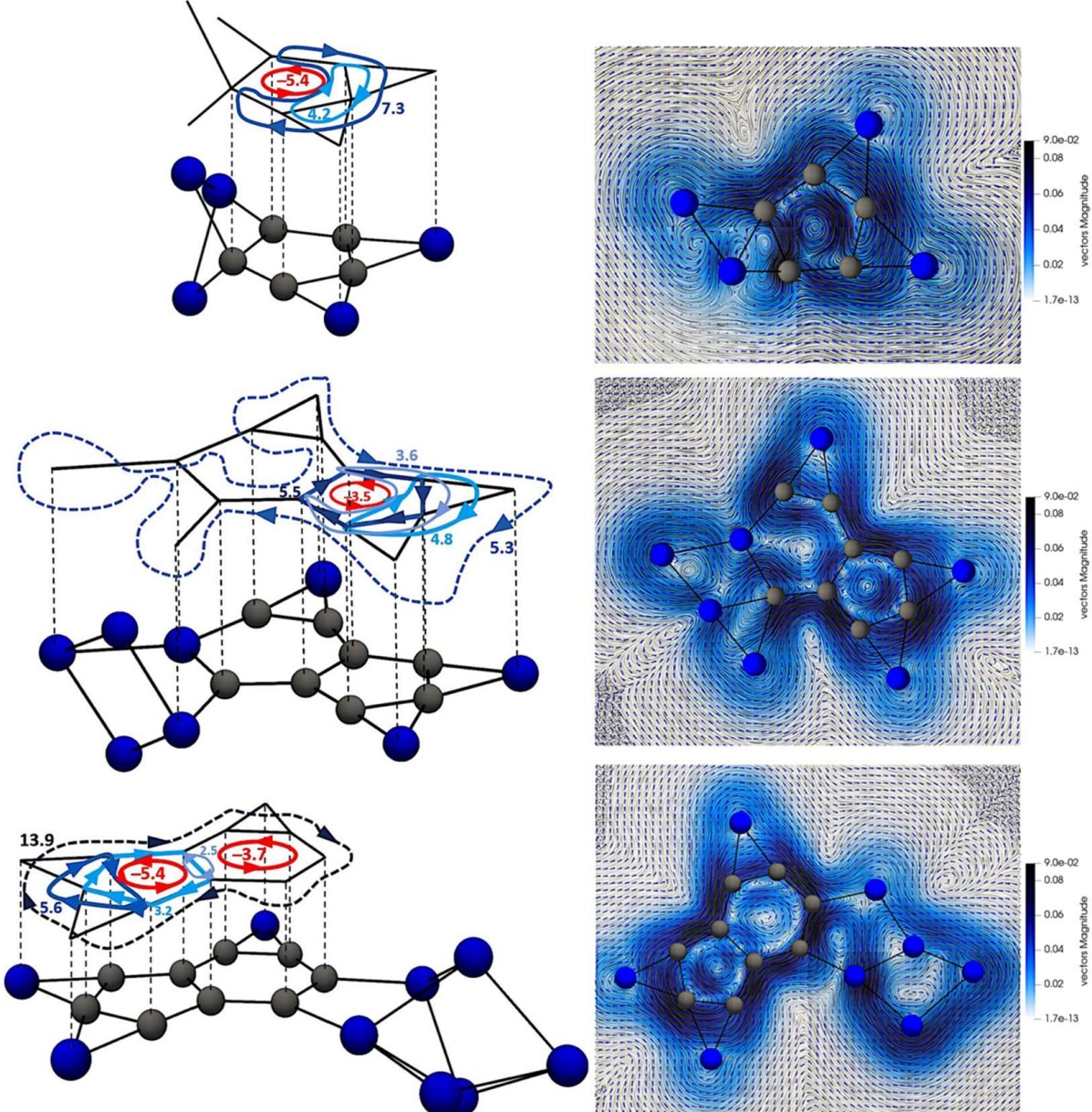

**Figure 6.** Schematic representation of local and global currents for $Si_5C_5$, $Si_8C_8$ and $Si_9C_9$ (**left**) and, the vector plot visualization of their current densities in a plane placed 0.5 Å above the molecular plane (**right**). RCS values (on each ring current circuit) and the vector intensities are in $nA \cdot T^{-1}$.

$Si_8C_8$ exhibits a local current around the pentagonal $C_5$ ring, which would result from the π bonds in this fragment. A horseshoe-shaped diatropic current circuit is also evident around the $C_3$–ptC–$C_2$ fragment. In addition, there is a semi-local diatropic current that does not close the $SiC_5$ hexagonal circuit; however, it does surround the $C_5$ ring. This is related to the delocalized π-bonds distributed throughout the system. $Si_9C_9$ exhibits a local diatropic current around the $C_5$ ring (3.2 $nA.T^{-1}$) with participation of the π-orbitals. In addition, an intense global diatropic current (13.9 $nA.T^{-1}$) is evident, which is mainly delocalized around the $C_9$ ring. To summarize (Figure 6), $Si_5C_5$, $Si_8C_8$, and $Si_9C_9$ exhibit local aromaticity (σ and π), with the participation of bridged Si and ptC and semi-local aromaticity (in the first two), and global aromaticity in the case of $Si_9C_9$, as a consequence of the π-electron cloud.

## 4. Conclusions

Here, we highlight that the global minima of the $Si_5C_5$, $Si_8C_8$, and $Si_9C_9$ clusters comprise a planar tetracoordinate carbon (ptC), a feature unnoticed in previous works. Chemical bonding analysis allows us to establish that $C_n$ rings are formed, monocyclic in the case of $Si_5C_5$, and bicyclic in the other two (in the case of $Si_8C_8$, a Si atom close one of the rings). A pentagonal $C_5$ ring is present in the latter, similar to what happens in the former. The ptC is embedded in this pentagonal ring in all cases, in which two Si form a 3c–2e Si–ptC–Si σ-bond. In addition, several delocalized π-bonds are detected. Therefore, the bonding analysis suggests the possibility of both local (σ) and semi-local or global (π) aromaticity. The latter is corroborated by the analysis of induced current density by an external magnetic field; i.e., the presence of local, semi-local, and global diatropic rings—a characteristic of aromatic compounds—is confirmed. In particular, in all cases, a local ring current is detected between the Si–ptC–Si fragment, which confirms this local aromatic σ-contribution. The other contributions would be of the π-type. These findings open up the possibility of identifying other Si–C combinations in which conditions are provided to favor the presence of ptCs; that is, to favor local, semi-local and global aromaticity in these species.

**Supplementary Materials:** The following supporting information can be downloaded at: https://www.mdpi.com/article/10.3390/atoms10010027/s1, Figures S1–S3: Global minimum and low-lying isomers of SinCn (n = 5, 8 and 9), their point group symmetries and spectroscopic states. Relative energies are shown in kcal.mol−1 at PBE0/def2-TZVP and PBE0-D3/def2-TZVP (in parentheses) levels, including zero-point energy (ZPE) corrections. Figure S4: Bond length in Å (black), natural charges (red), and Wiberg bond indices (blue) for (a) Si5C5, (b) Si8C8, and (c) Si9C9 global minimum at the PBE0/def2-TZVP level. Figures S5–S8: (a) Vector plot visualization of the current density of C6H6, Si5C5, Si8C8 and Si9C9 in a plane placed 0.5 Å above the molecular plane and top view of integration planes. The intensities of the diatropic currents are indicated in blue and the intensity of the paratropic currents is red. The intensity of the total current susceptibility is the sum of the paratropic and diatropic contributions. (b) Integration profiles along the integration planes of C6H6, Si5C5, Si8C8, and Si9C9. Table S1: Cartesian Coordinates of the $Si_5C_5$, $Si_8C_8$, $Si_9C_9$ global minimum calculated at the PBE0/def2-TZVP level of theory.

**Author Contributions:** Conceptualization, W.T., V.G., L.L.-P. and O.Y.; methodology, D.I., O.Y., J.C.C., V.G. and J.G.; software, D.I., O.Y., J.C.C. and J.G.; validation, W.T., O.Y. and V.G.; formal analysis, W.T. and L.L.-P.; investigation, W.T., L.L.-P. and O.Y.; resources, W.T. and J.G.; data curation, L.L.-P., V.G. and J.C.C.; writing—original draft preparation, W.T., L.L.-P. and V.G.; writing—review and editing, all authors.; visualization, W.T., O.Y. and V.G.; supervision, W.T.; project administration, W.T.; funding acquisition, W.T. and J.G. All authors have read and agreed to the published version of the manuscript.

**Funding:** This research was funded by FONDECYT grant number 1211128.

**Institutional Review Board Statement:** Not applicable.

**Informed Consent Statement:** Not applicable.

**Data Availability Statement:** Not applicable.

**Acknowledgments:** We thank the financial support of the National Agency for Research and Development (ANID) through FONDECYT project 1211128 (W.T.) and National Agency for Research and Development (ANID)/Scholarship Program/BECAS DOCTORADO NACIONAL/2019-21190427 (D.I.). National Agency for Research and Development (ANID)/Scholarship Program/BECAS DOCTORADO NACIONAL/2020-21201177 (L.L.-P.). Powered@NLHPC: This research was partially supported by the supercomputing infrastructure of the NLHPC (ECM-02).

**Conflicts of Interest:** The authors declare no conflict of interest.

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
