# Peer review of "Persistent Planar Tetracoordinate Carbon in Global Minima Structures of Silicon-Carbon Clusters"

_atoms, doi:10.3390/atoms10010027_

Round 1

Reviewer 1 Report

The authors present the global minima of the SixCx clusters comprised of planar tetracoordinate carbon and analyze the nature of chemical bonding with magnetic properties. The authors presented a very good introduction and used state-of-the-art methods for the calculations. This work is scientifically sound and the methodology valid. 

I have the following objections/criticisms and recommendations.

(1) The examined SixCx clusters are proposed to be a global minimum. They may be local minimum structures. Did they apply any atomic distortions to these clusters? Did the authors try all the possible positions and configurations to find the global minimum? The atomic structures may not maintain the presented geometry/configurations upon the small distortion via temperature.

(2) I can not distinguish some of the atomic bonds to be sigma, bond, or mix. They should address the bond nature in detail. 

(3) I strongly recommend the following references in the manuscript, including tetragonal and pentagonal SiC structures.

https://doi.org/10.1021/ja107711m

https://doi.org/10.1016/j.carbon.2020.12.003

Reviewer 2 Report

The paper by Leyva-Parra et. at. titled "Persistent Planar Tetracoordinate Carbon in Global Minima Structures of Silicon-Carbon Clusters" reports on density functional calculations on a series of Carbon-Silicon clusters containing a planar, tetra-coordinated carbon atom. The work reports on investigations on the molecular orbitals and, in particular, on the ring currents which can be expected from molecules showing, at least partial, aromatic characters.

The paper is written in an clear way, the results presented are consistent and overall quite important, for the general question on how aromaticity is formed. The conclusions drawn from the results are well justified.

The computational methods employed are ok for that kind of computations. My only concern, or question, is, if including dispersive corrections (like Grimme's D3 method or more advanced versions thereof) would affect the results. Maybe the authors could elucidate on that point. Otherwise the paper is pretty fine.

Round 2

Reviewer 1 Report

The authors addressed all my concerns.